# Proteomic and Metabolomic Evaluation of Insect- and Herbicide-Resistant Maize Seeds

**DOI:** 10.3390/metabo12111078

**Published:** 2022-11-07

**Authors:** Weixiao Liu, Lixia Meng, Weiling Zhao, Zhanchao Wang, Chaohua Miao, Yusong Wan, Wujun Jin

**Affiliations:** Biotechnology Research Institute, Chinese Academy of Agricultural Sciences, Beijing 100081, China

**Keywords:** safety evaluation, LFQ, widely targeted metabolomics, DEP, DAM, insect- and herbicide-resistant maize seeds

## Abstract

Label-free quantitative proteomic (LFQ) and widely targeted metabolomic analyses were applied in the safety evaluation of three genetically modified (GM) maize varieties, BBL, BFL-1, and BFL-2, in addition to their corresponding non-GM parent maize. A total of 76, 40, and 25 differentially expressed proteins (DEPs) were screened out in BBL, BFL-1, and BFL-2, respectively, and their abundance compared was with that in their non-GM parents. Kyoto Encyclopedia of Genes and Genomes (KEGG) pathway enrichment analysis showed that most of the DEPs participate in biosynthesis of secondary metabolites, biosynthesis of amino acids, and metabolic pathways. Metabolomic analyses revealed 145, 178, and 88 differentially accumulated metabolites (DAMs) in the BBL/ZH58, BFL-1/ZH58, and BFL-2/ZH58×CH72 comparisons, respectively. KEGG pathway enrichment analysis showed that most of the DAMs are involved in biosynthesis of amino acids, and in arginine and proline metabolism. Three co-DEPs and 11 co-DAMs were identified in the seeds of these GM maize lines. The proteomic profiling of seeds showed that the GM maize varieties were not dramatically different from their non-GM control. Similarly, the metabolomic profiling of seeds showed no dramatic changes in the GM/non-GM maize varieties compared with the GM/GM and non-GM/non-GM maize varieties. The genetic background of the transgenic maize was found to have some influence on its proteomic and metabolomic profiles.

## 1. Introduction

Since the large-scale commercialization of GM crops began in 1996, the corresponding planting area has expanded rapidly worldwide while species and transgenic traits of GM crops have been constantly enriched. The planting area of GM crops with multiple transgenic traits continues to increase, accounting for 42% of GM crops worldwide [1]. Insect and herbicide resistance are two important transgenic traits that not only reduce the amount of applied chemical pesticides but also reduce labor costs. Despite the benefits associated with the commercialization of GM crops, their safety remains a focal point of concern [2,3]. Therefore, it is urgent that a systematic and comprehensive safety evaluation of GM crops be conducted [4,5,6,7].

Previously employed directional detection methods, such as polymerase chain reaction (PCR) and enzyme-linked immunosorbent assay (ELISA), specifically and accurately detect exogenous genes and proteins. Newly developed nondirectional detection methods, such as omics and especially proteomics and metabolomics, detect proteins and metabolites very closely related to the endpoint phenotypes [8], making it possible to provide overall insights into both unintended and intended changes in the analyzed GM crops [9,10]. Isobaric tags for relative and absolute quantitation (iTRAQ)-based proteomics are widely used in the safety evaluation of GM crops [11,12,13,14,15,16], however, expensive isotope-labeling reagents are required. LFQ proteomics is becoming increasingly popular due to its easy sample preparation, low protein loss, and no limitation of samples. In addition to proteins, metabolites also profoundly influence every aspect of the growth and development of crops [17,18]. The advances in metabolomics make it is possible to compare, in detail, the metabolic profiles between GM crops and their non-GM counterparts [19,20,21].

The insect resistance and herbicide resistance of GM crops are the main and popular features [22]. The *cry* gene, encoding Cry protein showing a broad insecticidal spectrum, has been widely used [23]. As for herbicide resistance genes, the *epsps* gene, encoding 5-enolpyruvate shikimic acid-3-phosphate synthase (EPSPS), is most widely used in GM crops [24,25,26]. Previous studies have focused more on environmental factors and the impact of GM on crop growth processes. In this study, LFQ proteomic and widely targeted metabolomic analyses were applied to evaluate the unintended and intended changes in the GM maize seeds of varieties BBL (carrying the *cry3Bb, cry1Ab*, and *cp4-epsps* genes), BFL-1, and BFL-2 (carrying the *cry1F*, *cry1Ab*, and *cp4-epsps* genes). These selected GM maize varieties show high insect resistance and excellent herbicide resistance, which has good application prospects. As storage and edible tissues, seeds contain abundant protein and metabolites that are more suitable and necessary to be studied. Based on these omics data, the profile changes of the proteins and metabolites will be elucidated and compared between three GM maize varieties and their non-GM parents, between GM maize varieties, and between non-GM maize varieties. It is conducive to promoting the development, seed production and preservation of these GM maize varieties, and the stable promotion of these GM maize industrialization.

## 2. Materials and Methods

### 2.1. Plant Materials

Seeds of the GM maize varieties BBL, BFL-1 and BFL-2, and their non-GM parents, ZH58 (with strong resistance to drought and various diseases such as powdery mildew) and ZH58×CH72 (with high resistance and lodging, uniform ears, and seed yield up to 90%), were studied. The information of the studied maize varieties are listed in Table 1.

### 2.2. DNA Extraction and PCR-Based Detection of GM Maize

A genomic DNA extraction kit (Tiangen, Beijing, China) was used to prepare genomic DNA from maize seeds. Event- and gene-specific PCR were applied to confirm the identity of the GM maize varieties based on genotype. The sequences of the primers used and the PCR product sizes are listed in Appendix A.

### 2.3. Protein Preparation and Trypsin Digestion

Maize seeds were ground in liquid nitrogen and incubated in lysis buffer, then reduced with10 mM Dimercapterol (DTT) (Sigma, Shanghai, China). The suspension was sonicated on ice and centrifuged. The protein was precipitated by precooled acetone. After centrifugation, the protein pellets were air-dried and resuspended in 100 mM tetraethylammonium bromide (TEAB) (Sigma, Shanghai, China) pH 8.0 containing 8M urea. The protein samples were reduced with 10 mM DTT and then alkylated with 50 mM iodoacetamide (IAM) (Sigma, Shanghai, China). After repeatedly precipitated by acetone, centrifugated, air-dried and resuspended in 100 mM TEAB (pH 8.0) containing 8M urea (Sigma, Shanghai, China), the total protein concentration was measured using the Bradford method [27].

Trypsin (MS grade, Sigma, Shanghai, China) and protein from each sample were mixed at ratio of 1:50 (*w*/*w*). Digestion was performed at 37 °C for 16 h. After digestion, the peptides were desalted using C18 columns and dried with a vacuum concentration. Three biological replicates of the seeds of different maize varieties were used for proteomic profiling.

### 2.4. LC–MS/MS Analysis

The dried peptide sample was reconstituted with a 0.1% formic acid (FA) (HPLC grade, Sigma, Shanghai, China) aqueous solution and then centrifuged at 15,000 rpm for 10 min. Then, the sample solution was analyzed by HPLC–MS. The detailed analysis procedure is presented in the Appendix A.

### 2.5. Metabolite Preparation

Maize seed grains of each variety were ground in liquid N2. Total metabolites were extracted with 70% aqueous methanol (HPLC grade, Sigma, Shanghai, China) at 4 °C. Following centrifugation at 10,000× *g* for 10 min, the extracts were absorbed using a CNWBOND Carbon-GCB SPE Cartridge (ANPEL, Shanghai, China) and filtered before UPLC–MS/MS analysis. Six biological replicates of the seeds of different maize varieties were used for metabolic profiling.

### 2.6. UPLC Conditions and ESI-Q TRAP-MS/MS

The extracted metabolites were analyzed using an UPLC–ESI-MS/MS system (UPLC, Shim-pack UFLC SHIMADZU CBM30A system; MS, Applied Biosystems 4500 Q TRAP). The analytical conditions were presented in the Appendix A.

### 2.7. Data Analysis

Protein identification was performed against the UniProt Zea mays (maize) database supplemented with four foreign proteins, EPSPS, Cry1Ab, Cry3Bb, and Cry1F. All identified proteins were matched with at least one unique peptide at ≥95% confidence [12,28]. Proteins were considered DEPs based on ≥2 or ≤0.5 fold change and *p* ≤ 0.05 [13,29,30] in the comparison groups. The *p*-value and fold change values of the univariate analysis were combined to further screen differential metabolites. Metabolites with ≥2 or ≤0.5 fold change and VIP ≥ 1 were considered differential metabolites for group discrimination [31,32]. A heatmap based on the hierarchical cluster analysis method was generated by Genesis software. The principal components analysis (PCA) was generated in R software (www.r-project.org, R Version1.6.2 4.0.2). KEGG pathway enrichment analysis of the DEPs and DAMs was carried out using the KEGG database (http://www.genome.jp/kegg/, released on 1 May 2017) [33,34].

### 2.8. ELISA

Maize seeds were ground in liquid N2. Total proteins were extracted with lysis buffer using an ELISA kit. The contents of foreign proteins Cry1Ab, Cry1F, Cry3Bb, and EPSPS were measured using ELISA kits according to the manufacturer’s instructions (Youlong, Shanghai, China).

## 3. Results

### 3.1. Identification of GM Maize Lines

The proteomic and metabolomic profiles of three selected GM maize varieties BBL, BFL-1, and BFL-2 and their non-GM parents ZH58 and ZH58×CH72 were studied, and their identities were first confirmed by event- and gene-specific PCR. Specific PCR fragments were only obtained from three GM maize varieties, as expected (Figure 1A).

### 3.2. Protein and Metabolic Profiling of Maize Seeds

A total of 3577 proteins were successfully detected in the maize seeds using LFQ proteomics. The principal component (PC) analysis led to PC1 (50.60%) and PC2 (18.21%) being obtained as the two main components (Figure 1B). Cluster analysis of the identified proteins showed that BBL and BFL-1, and BFL-2 and ZH58×CH72 shared the highest similarity among the five studied maize varieties based on patterns of protein abundance. BFL-2 and ZH58 shared higher similarity than BFL-1 and ZH58 on the basis of trends in protein abundances (Figure 1C).

A total of 704 metabolites were successfully detected in the maize seeds using widely targeted metabolomics. The detected metabolites were diverse and could be classified into 12 classes, namely alkaloids (15.48%), amino acids and derivatives (10.65%), flavonoids (11.22%), lignans and coumarins (2.41%), lipids (18.89%), nucleotides and derivatives (5.97%), organic acids (6.96%), phenolic acids (12.36%), quinones (0.57%), tannins (0.14%), terpenoids (3.12%), and others (12.22%) (Figure 2A). The principal component (PC) analysis led to PC1 (35.11%) and PC2 (15.68%) being obtained as the two main components (Figure 2B). Cluster analysis showed that the five studied maize varieties clustered into two groups. First, based on metabolic profiling, BBL and BFL-1 shared the highest similarity among the five studied maize lines, and BBL and ZH58 shared higher similarity than BFL-1 and ZH58. Second, the metabolic profiles of BFL-2 and ZH58×CH72 shared higher similarity than those of BFL-1 and BFL-2 (Figure 2C).

### 3.3. DEPs Detection in Maize Seeds by LFQ Proteomic Analysis

The number and regulation state of the DEPs in the different comparison groups are summarized in Table 2. There were 47 upregulated proteins and 29 downregulated proteins, corresponding to a total of 76 DEPs identified in the BBL/ZH58 comparison group (Appendix A). Forty DEPs were identified via comparison of BFL-1 with ZH58, with half being upregulated and half being downregulated (Appendix A). A total of 25 DEPs were identified in the BFL-2/ZH58×CH72 comparison group, including 19 upregulated proteins and 6 downregulated proteins (Appendix A). Additionally, 42 and 30 DEPs identified in the BFL-1/BFL-2 and ZH58/ZH58×CH72 comparison groups, respectively (list of DEPs not shown).

### 3.4. KEGG Pathway Enrichment Analysis of the Identified DEPs

KEGG pathway enrichment analysis showed that the DEPs identified in the BBL/ZH58 comparison group are mainly involved in biosynthesis of secondary metabolites (ko01110) and biosynthesis of amino acids (ko01230), followed by diterpenoid biosynthesis (ko00904), amino sugar and nucleotide sugar metabolism (ko00520), and 2-oxocarboxylic acid metabolism (ko01210) (Figure 3A). The DEPs obtained from the BFL-1/ZH58 comparison group mainly participate in metabolic pathways (ko01100) and biosynthesis of amino acids (ko01230), followed by carbon metabolism (ko01200) (Figure 3B). The DEPs obtained from the BFL-2/ZH58×CH72 comparison group mainly participate in metabolic pathways (ko01100) and biosynthesis of secondary metabolites (ko01110), followed by protein processing in endoplasmic reticulum (ko04141) (Figure 3C).

### 3.5. DAMs Detection in Maize Seeds by Widely Targeted Metabolomic Analysis

The number and regulation state of DAMs in the different comparison groups are summarized in Table 3. There were 66 increased and 79 decreased metabolites, totaling 145 DAMs identified in the BBL/ZH58 comparison group (Appendix A). A total of 178 DAMs were identified via comparison of BFL-1 with ZH58, 18 of which were increased and 160 of which were decreased (Appendix A). A total of 88 DAMs were identified in the BFL-2/ZH58×CH72 comparison group, including 72 increased metabolites and 16 decreased metabolites (Appendix A). Additionally, 189 and 286 DAMs were identified in the BFL-2/BFL-1 and ZH58×CH72/ZH58 comparison groups, respectively (list of DAMs not shown).

### 3.6. KEGG Pathway Enrichment Analysis of the Identified DAMs

The DAMs identified in the BBL/ZH58 comparison group are mainly involved in ABC transporters (ko02010), followed by biosynthesis of amino acids (ko01230) and arginine and proline metabolism (ko00330) (Figure 4A). The DAMs obtained from the BFL-1/ZH58 comparison group mainly participate in biosynthesis of amino acids (ko01230) and carbon metabolism (ko01200), followed by glyoxylate and dicarboxylate metabolism (ko00630) (Figure 4B). The DAMs detected in the BFL-2/ZH58×CH72 comparison group mainly participate in metabolic pathways (ko01100) and arginine and proline metabolism (ko00330), followed by lysine degradation (ko00310) (Figure 4C).

### 3.7. Identification of co-DEPs and co-DAMs in the Seeds

Three DEPs, namely, EPSPS, B4FR99, and A0A804MXV9, were simultaneously identified in the three comparison groups and named co-DEPs (Figure 5A). These co-DEPs were unregulated among the three comparison groups (Table 4). Among the DAMs, 11 were identified in three comparison groups and named co-DAMs (Figure 5B). The regulatory trend of these co-DAMs is shown in Table 5. Three co-DAMs, namely mws0520, Smpn009074, and Lmsn009824, were upregulated, but one co-DAM, pmn001319, was downregulated in the three GM/non-GM comparison groups. Lmlp003161, pme1173, pme2693, mws0005, mws0133, and Lmmp002013 were downregulated in BBL/ZH58 and BFL-1/ZH58 but upregulated in BFL-2/ZH58×CH72. Pme2914 was upregulated in BBL/ZH58 and BFL-2/ZH58×CH72 but downregulated in BFL-1/ZH58.

### 3.8. Integrated Proteomic and Metabolomic Analyses

Integrated proteomic and metabolomic analyses showed that the DEPs and DAMs in BBL/ZH58 (Figure 6A), BFL-1/ZH58 (Figure 6B), and BFL-2/ZH58×CH72 (Figure 6C) are mainly involved in KEGG pathway metabolic pathways (ko01100), biosynthesis of secondary metabolites (ko01110), and biosynthesis of amino acids (ko01230).

### 3.9. Exogenous Protein Detection by ELISA and LFQ Proteomics of Seeds of GM Maize Varieties

The ELISA data showed that the contents of three exogenous proteins Cry1Ab, Cry3Bb, and EPSPS were 3.35 ± 0.24 μg/g, 2.26 ± 0.01 μg/g, and 20.86 ± 3.41 μg/g in the BBL maize seeds, respectively; the contents of Cry1Ab, Cry1F, and EPSPS were 1.95 ± 0.01 μg/g, 39.46 ± 0.22 μg/g, and 3.79 ± 0.08 μg/g, respectively, in BFL-1 and 1.81 ± 0.07 μg/g, 33.72 ± 0.71 μg/g, and 3.49 ± 0.26 μg/g, respectively, in BFL-2. The proteomics data showed that Cry1Ab and EPSPS were identified as DEPs in BBL/ZH58, and Cry1F and EPSPS were detected as DEPs in BFL-2/ZH58×CH72, but only EPSPS was identified as a DEP in BFL-1/ZH58 (Table 6).

## 4. Discussion

Different from previous studies focusing on non-edible tissues playing important roles during crop growth [35,36,37,38] or the influence of growing environmental factors [39,40,41,42], the need for in-depth systematic research on the safety of maize seeds as an important feed and edible material rich in protein and metabolites, is becoming a matter of urgency. In this study, LFQ proteomic and widely targeted metabolomic analyses were performed on three GM maize varieties, namely, BBL, BFL-1, and BFL-2, and their corresponding non-GM parents ZH58 and ZH58×CH72, all grown in the same environment. Of these, BFL-1 and BFL-2 carry the same exogenous genes (*cry1Ab*, *cry1F*, and *cp4-epsps*) but have the different genetic backgrounds, ZH58 and ZH58×CH72, respectively. In addition to the exogenous genes *cry1Ab* and *cp4-epsps*, BBL and BFL-1 carry the *cry3Bb* and *cry1F* genes, respectively, but have the same genetic background ZH58. Such an experimental design allowed us to analyze, compare, and mine the data from multiple perspectives.

There were 76, 40, and 25 maize DEPs detected in the BBL/ZH58, BFL-1/ZH58, and BFL-2/ZH58×CH72 comparisons, respectively. A total of 145, 178, and 88 DAMs were found in the BBL/ZH58, BFL-1/ZH58, and BFL-2/ZH58×CH72 comparisons, respectively. None of these DEPs and DAMs were identified as novel toxins or allergens, only changed in abundance, consistent with previous reports [4,24,43]. The GM maize varieties were not dramatically different to their non-GM parents based on seed proteomic profiles; only less than 2.5% of the total identified proteins were screened out as DEPs. Based on metabolomic profiling of seeds, the GM/non-GM (145, 178, and 88 DAMs) maize varieties were not dramatically changed compared with the GM/GM (189 DAMs) and non-GM/non-GM (286 DAMs) maize varieties. Compared with GM maize BBL and BFL-1, BFL-2 did not have significantly different protein and metabolic profiles. This suggests that genetic background has some influence on changes in the proteins and metabolites in transgenic crops. In these studied GM maize varieties, the *cry1Ab*, *cry1F*, and *cry3Bb* are *Bt* genes are exogenous, and the insecticidal proteins Cry1Ab and Cry1F are not involved in any known metabolic activity in plants. Cry3Bb was not even identified as a DEP, which may be due to methodological limitations. Herbicide tolerance is conferred by introducing the gene *cp4-epsps*, which encodes CP4-EPSPS, a key enzyme in the shikimic acid pathway. EPSPS is involved in the metabolic pathway for the biosynthesis of aromatic amino acids in microorganisms and plants [44,45,46]. Nevertheless, the introduction of EPSPS did not affect the levels of other enzymes in the shikimic acid pathway in seeds of the three GM maize varieties (Appendix A). Integrated proteomic and metabolomic analyses can better reveal the biological processes of GM crops. The introduction of EPSPS was not found to cause any changes in the abundance of metabolites involved in the shikimic acid pathway in the seeds of the GM maize varieties BBL and BFL-2; only the abundance of quinate decreased in the seeds of the GM maize variety BFL-1 (Appendix A). This could be due to the location of *cp4-epsps* gene insertion in genomic DNA being different in the various GM maize varieties, thus affecting different biological processes, or because of the influence of different genetic backgrounds.

Three co-DEPs and eleven co-DAMs were identified in the seeds of the three GM maize varieties. The alignment of these proteins and metabolites to those screened out in previous studies is expected to form the basis for identifying screening markers of GM crops if they are frequently present in transgenic crops. Of course, this is dependent on sufficient research to accumulate more alternative indicators in backup databases to ensure reliability.

## 5. Conclusions

The proteomic profiling of the GM maize variety seeds were not dramatically changed compared with those of their non-GM control. The metabolomic profiling of the GM/non-GM maize variety seeds were not dramatically changed compared with those of the GM/GM and non-GM/non-GM maize variety seeds. The genetic background of transgenic maize also has some influence on its proteomic and metabolomic profiling.

## Figures and Tables

**Figure 1 metabolites-12-01078-f001:**
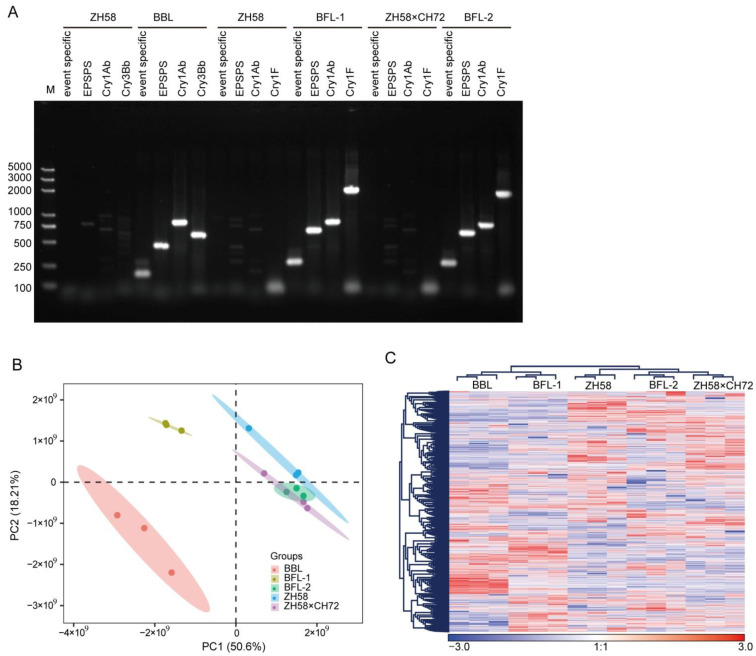
Proteomic profiling of seeds of the studied maize varieties. (**A**) Event- and gene-specific PCR-based detection of the studied GM maize varieties. M, Trans2k plus marker. (**B**) Principal component (PC) analyses of the protein levels in the seeds of five maize varieties. Score plot of the first two PCs with the explained variance. (**C**) Cluster map comparing the DEP regulation patterns. Red indicates relatively high expression, blue indicates relatively low expression, and white indicates the same expression levels in the two lines. All MS data were normalized and then applied in cluster analysis.

**Figure 2 metabolites-12-01078-f002:**
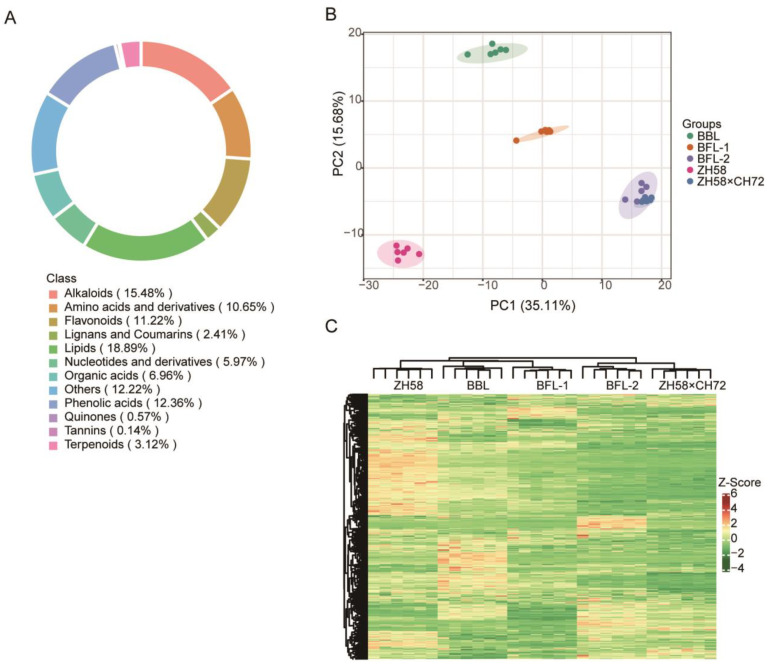
Metabolomic profiling of seeds of the studied maize varieties. (**A**) Circular plot of the metabolite class composition. (**B**) Principal component (PC) analyses of the amount of metabolite in the seeds of five maize varieties. Score plot of the first two PCs with the explained variance. (**C**) Hierarchical clustering of five maize lines using metabolite accumulation data. In the heatmap, all replicates of each maize line are visualized in a single column. Metabolite accumulation is shown in different colors, where red indicates high abundance and green indicates low relative expression.

**Figure 3 metabolites-12-01078-f003:**
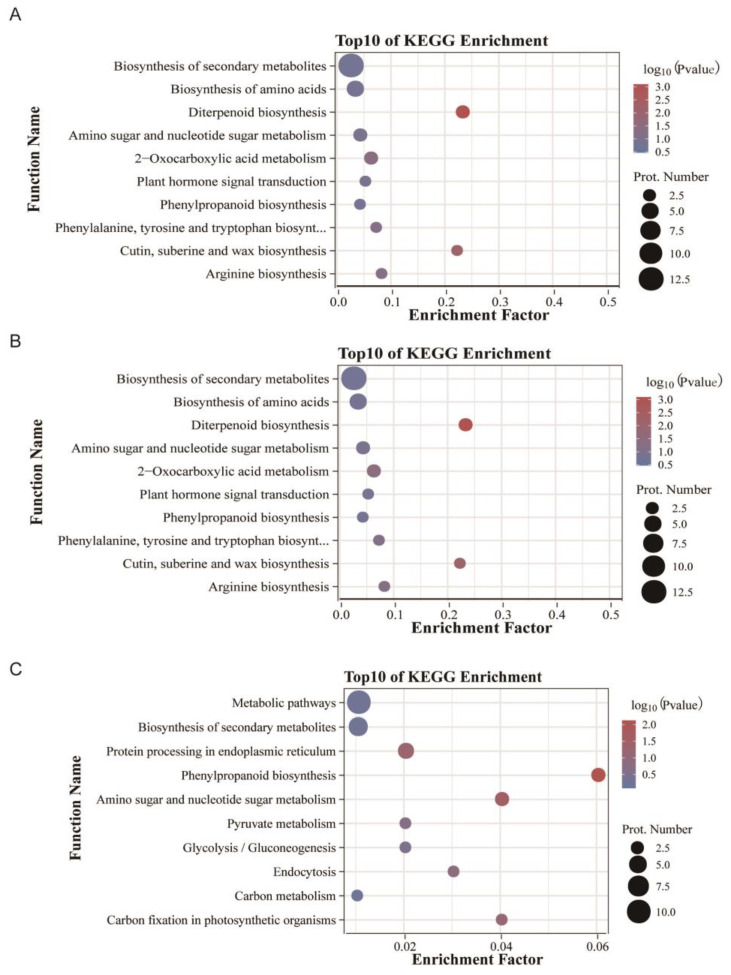
KEGG pathway enrichment analysis of DEPs in BBL/ZH58 (**A**), BFL-1/ZH58 (**B**), and BFL-2/ZH58×CH72 (**C**).

**Figure 4 metabolites-12-01078-f004:**
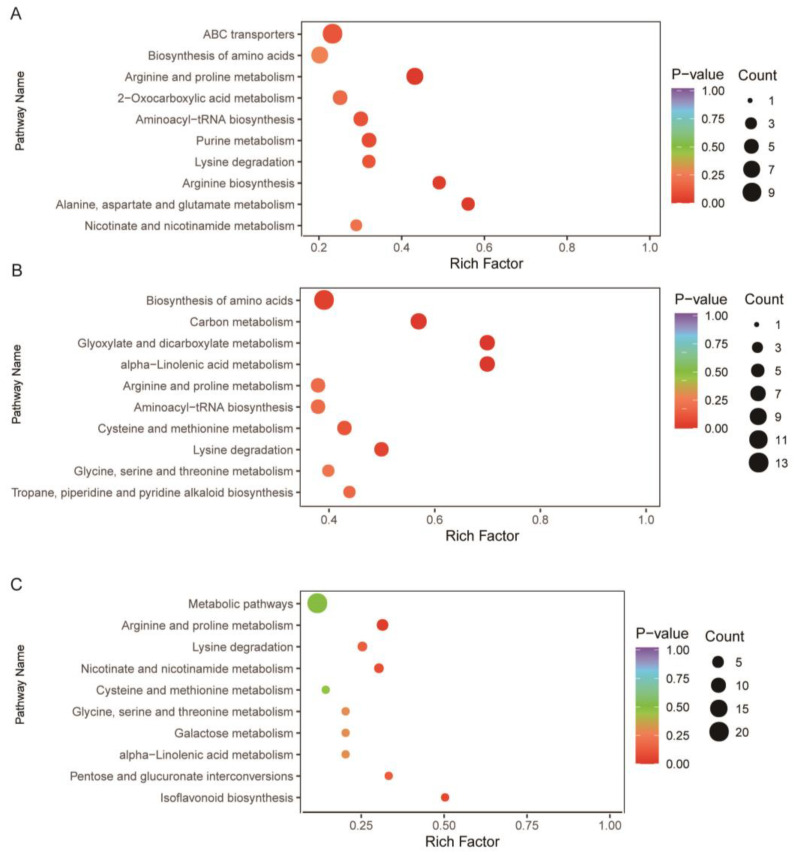
KEGG pathway enrichment analysis of DAMs in BBL/ZH58 (**A**), BFL-1/ZH58 (**B**), and BFL-2/ ZH58×CH72 (**C**).

**Figure 5 metabolites-12-01078-f005:**
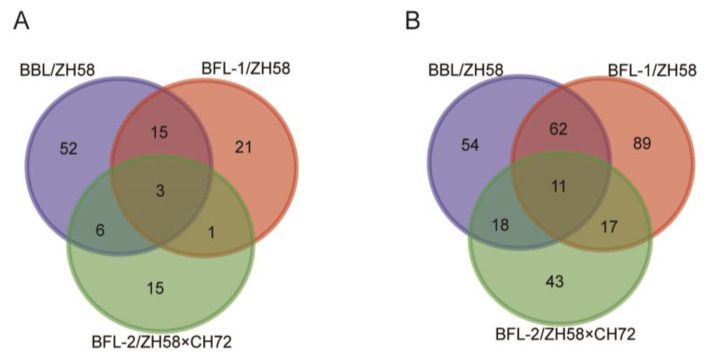
Venn diagram showing the number of overlapping DEPs (**A**) and DAMs (**B**) in pairwise comparisons between three groups of GM/non-GM maize seeds.

**Figure 6 metabolites-12-01078-f006:**
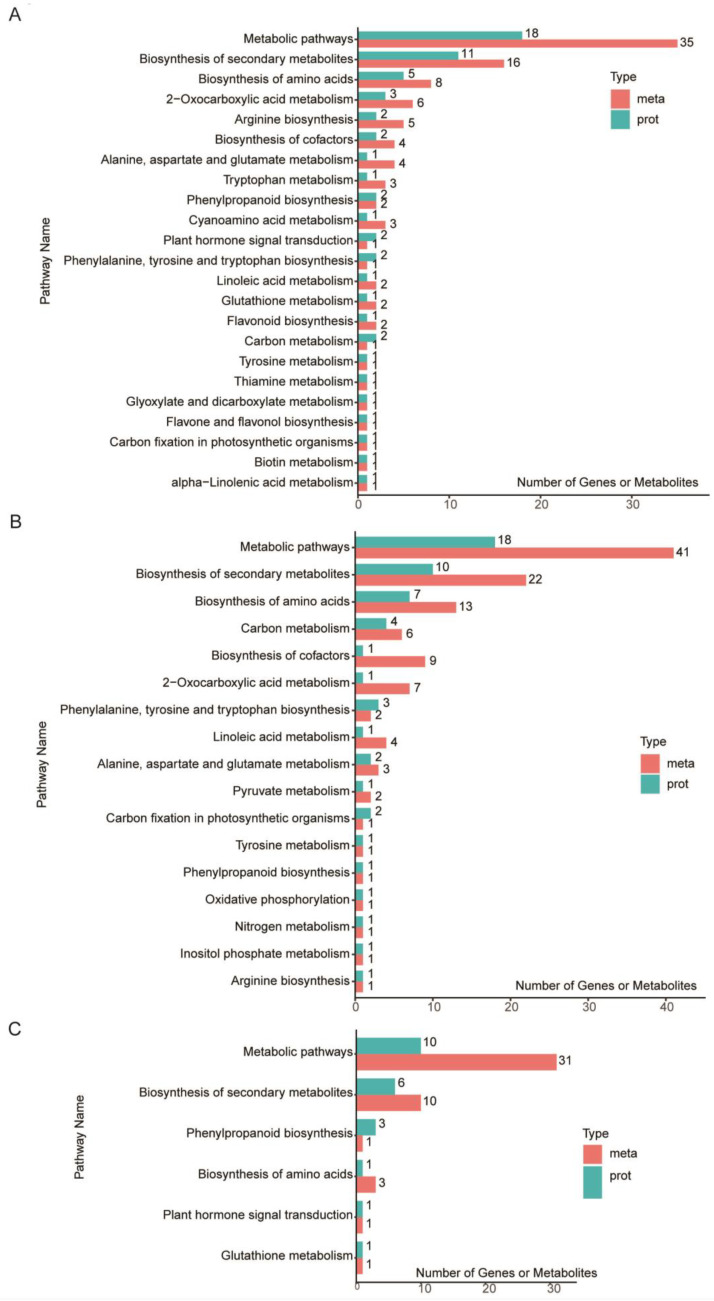
KEGG pathway enrichment analysis of the integrated DEPs and DAMs in BBL/ZH58 (**A**), BFL-1/ZH58 (**B**), and BFL-2/ZH58×CH72 (**C**).

**Table 1 metabolites-12-01078-t001:** Summary of the studied maize lines.

Natural Genotypic Maize Lines	GM Maize Lines	Foreign Proteins
ZH58	BBL	EPSPS
		Cry1Ab
		Cry3Bb
	BFL-1	EPSPS
		Cry1Ab
		Cry1F
ZH58×CH72	BFL-2	EPSPS
		Cry1Ab
		Cry1F

**Table 2 metabolites-12-01078-t002:** Summary of the number and regulated state of the DEPs in the different comparison groups.

Comparison Groups	No. of Upregulated Proteins	No. of Downregulated Proteins	No. of DEPs
**BBL/ZH58**	47	29	76
**BFL-1/ZH58**	20	20	40
**BFL-2/ZH58×CH72**	19	6	25
**BFL-1/BFL-2**			42
**ZH58/ZH58×CH72**			30

**Table 3 metabolites-12-01078-t003:** Summary of the number of DAMs in the different comparison groups.

Comparison Groups	No. of Upregulated Metabolites	No. of Downregulated Metabolites	No. of DAMs
**BBL/ZH58**	66	79	145
**BFL-1/ZH58**	18	160	178
**BFL-2/ZH58×CH72**	72	16	88
**BFL-2/BFL-1**			189
**ZH58×CH72/ZH58**			286

**Table 4 metabolites-12-01078-t004:** The DEPs among the three GM maize lines.

Accession	Name	State	KEGG Enrichment Pathway
	EPSPS	Up	Phenylalanine, tyrosine, and tryptophan biosynthesis (ko00400); Metabolic pathways (ko01100); Biosynthesis of secondary metabolites (ko01110); Biosynthesis of amino acids (ko01230)
**B4FR99**	Acidic endochitinase	Up	Amino sugar and nucleotide sugar metabolism (ko00520); Metabolic pathways (ko01100)
**A0A804MXV9**	Uncharacterized protein	Up	-

-, not annotated to KEGG pathway.

**Table 5 metabolites-12-01078-t005:** The DAMs among the three GM maize lines.

Accession	Name	BBL	BFL-1	BFL-2	KEGG Enrichment Pathway
**mws0520**	N-Acetyl-L-tyrosine	Up	Up	Up	-
**Smpn009074**	2α,3α,19α,23-tetrahydroxy-12-ursen-28-oic acid	Up	Up	Up	-
**Lmsn009824**	2α,3β,19α,23-Tetrahydroxyolean-12-en-28-oic acid	Up	Up	Up	-
**pmn001319**	1-O-Feruloyl-3-O-p-Coumaroylglycerol	Down	Down	Down	-
**pme1173**	Allopurinol	Down	Down	Up	-
**Lmlp003161**	N-Feruloylputrescine	Down	Down	Up	Arginine and proline metabolism (ko00330); Metabolic pathways (ko01100)
**pme2693**	N-Acetylputrescine	Down	Down	Up	Arginine and proline metabolism (ko00330); Metabolic pathways (ko01100)
**mws0005**	Tryptamine	Down	Down	Up	Tryptophan metabolism (ko00380); Indole alkaloid biosynthesis (ko00901); Metabolic pathways (ko01100); Biosynthesis of secondary metabolites (ko01110)
**mws0133**	Nicotinamide	Down	Down	Up	Nicotinate and nicotinamide metabolism (ko00760); Metabolic pathways (ko01100); Biosynthesis of cofactors (ko01240)
**Lmmp002013**	Dihydroferuloylputrescine	Down	Down	Up	-
**pme2914**	3-Hydroxy-3-methylpentane-1,5-dioic acid	Up	Down	Up	-

-, not annotated to KEGG pathway.

**Table 6 metabolites-12-01078-t006:** Exogenous proteins detected by LFQ proteomics and ELISA.

Comparison Group	Exogenous Proteins	Fold Change	*p*-Value	Exogenous Protein Content (μg/g) inGM Maize Seeds
**BBL/ZH58**	Cry1Ab	3.199	0.000658169	3.35 ± 0.24
	Cry3Bb	-	-	2.26 ± 0.01
	CP4-EPSPS	172.77	1.46348 × 10^−6^	20.86 ± 3.41
**BFL-1/ZH58**	Cry1Ab	-	-	1.95 ± 0.01
	Cry1F	-	-	39.46 ± 0.22
	CP4-EPSPS	23.718	0.000121842	3.79 ± 0.08
**BFL-2/ZH58×CH72**	Cry1Ab	-	-	1.81 ± 0.07
	Cry1F	26.586	0.004223503	33.72 ± 0.71
	CP4-EPSPS	16.396	1.3585 × 10^−7^	3.49 ± 0.26

-, values are not within the specified range (FC ≥ 2 or ≤0.5, *p* ≤ 0.05).

## Data Availability

Not applicable.

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
