# Peer review of "Proteomic and Metabolomic Evaluation of Insect- and Herbicide-Resistant Maize Seeds"

_metabolites, 2022, doi:10.3390/metabo12111078_

Round 1

Reviewer 1 Report

The manuscript in reference describes proteomics and metabolomics-based characterization of three genetically-modified maize varieties and their corresponding non-genetically modified parent maize. Although the manuscript has interesting results, some points should be addressed before further consideration.

1.       Detailed scrutiny should be performed throughout the manuscript to revise some grammar and stylistic issues.

2.       A clear justification for performing the present study on the test maize varieties should be provided since the reason mentioned by the authors is very general. In other words, the aim and scope of this study are not adequately structured and informed. For instance, no apparent reason is provided to characterize these three test maize varieties (i.e., BBL, BFL-1, and BFL-2) using proteomic and metabolomics-based analysis. Which are the hypothesis and the research question? This fact is exposed in the discussion section but is very laconic. The results are generally interpreted, but a more profound discussion should be reached.

3.      Based on the study hypothesis, what is the scope of the findings based on the detected DAMs, e.g., amino acids, phenolics, or substituted polyamines? Is there any influence on the traits of the studied GMOs? Is there any concrete relationship between proteins and these metabolites on phenotype varieties?

4.       The manuscript did not include a Conclusions section.

5.       Some abbreviations are not defined in the first use to facilitate interpretation for readers. Be consistent throughout the manuscript.

6.       Details regarding traits and phenotypes of the test varieties are not provided. The parent plant is not informed in detail.

7.       There is no clear differentiation between protein and metabolite findings. In addition, the scope of such findings is not clearly provided. In my opinion, section 3.7 is incompletely exploited.

8.       My main criticism of this manuscript is related to the metabolite identification since it is not adequately performed and even informed in the manuscript. Tables S5-S7 comprise extensive lists of metabolites, but nothing about their identification is provided. In this sense, HRMS is not entirely suitable for performing reliable compound identification because of the isomer occurrence. For instance, Table S7 includes the compounds Jmzn006005 (C10H10O3, 3,4-methylenedioxy cinnamyl alcohol) and mws1200 (C10H10O3, trans-4-Hydroxycinnamic Acid Methyl Ester), which are related to isomers. How can the authors ensure that these compounds are well-identified and are not false positives or misidentifications? If they are not correctly identified, how valuable and accurate is the information given by the authors for this special issue? Other examples are also in these tables, which radically limits the reliability of results.

9.       In general, M&M section requires deeper scrutiny since some experimental details are missing to ensure outcome reproducibility (even the information provided in supplementary material). For instance, the brand, model, and grade of reagents, solvents, materials, and instruments must be provided.

10.   The information provided in Figures 1, 2, and 3 is challenging to be adequately visualized.

Reviewer 2 Report

This research article is well described the  metabolomics and proteomics difference between genetically modified maize and non-genetically modified maize. The study is very interesting and explaining the difference techniques to evaluate the metabolic profiling. My comments are as follow to improve the quality of research article ;

1.     The abstract section is not unified, author should normalize the introduction section. For example abbreviation is not properly followed by authors.

2.     Keywords seems to go the out of scope. Authors should rethink about it. For example, maize seed and insect-and herbicide-resistant maize can be combined as insect-and herbicide-resistant maize seeds.

3.     The introduction section is poorly written. It does not reflect the novelty of current research and previous research regarding  in this fields.

4.     The term such as DDT, TEAB need abbreviation.

5.     The result are not well explained. Need more explanation in result section.  Some data is too much complicated to understand. Authors should simplified it.

Round 2

Reviewer 2 Report

article has been modified as per reviewer comments and improved the quality. So, I think we can published the manuscript in the present form.